# Diabetes Is an Independent Risk Factor for Cancer after Heart and/or Lung Transplantation

**DOI:** 10.3390/jcm11144127

**Published:** 2022-07-15

**Authors:** Hristo Kirov, Alexandros Moschovas, Tulio Caldonazo, Imke Schwan, Gloria Faerber, Tim Sandhaus, Thomas Lehmann, Torsten Doenst

**Affiliations:** 1Department of Cardiothoracic Surgery, Jena University Hospital, 07743 Jena, Germany; hristo.kirov@med.uni-jena.de (H.K.); alexandros.moschovas@med.uni-jena.de (A.M.); tulio.caldonazo@med.uni-jena.de (T.C.); imke.schwan@med.uni-jena.de (I.S.); gloria.faerber@med.uni-jena.de (G.F.); tim.sandhaus@med.uni-jena.de (T.S.); 2Institute of Medical Statistics, Computer and Data Sciences, Jena University Hospital, 07743 Jena, Germany; thomas.lehmann@med.uni-jena.de

**Keywords:** heart transplantation, lung transplantation, diabetes mellitus, cancer

## Abstract

Background: De novo cancers are feared complications after heart or lung transplantation. Recent data suggest that diabetes mellitus (DM) might also be a risk factor for cancer. We hypothesized that transplanted diabetic patients are at greater risk of developing cancer compared to non-diabetic ones. Methods: We reviewed 353 patients post-heart and/or -lung transplantation from our center between October 1999 and June 2021. Patients with follow-up <180 days (*n* = 87) were excluded from the analysis. The remaining 266 patients were divided into patients who had preoperative DM *(n* = 88) or developed it during follow-up (*n* = 40) and patients without DM (*n* = 138). Results: The diabetic cohort showed higher rates of malignancies in all patients (30.33 vs. 15.97%, *p* = 0.005) and in the matched population (31.9 vs. 16.1%, *p* < 0.001). There were also significantly more solid tumors (17.9 vs. 9.4%, *p* = 0.042; matched: 16.6 vs. 9.1%, *p* = 0.09) The presence of diabetes was associated with a 13% increased risk of cancer when compared to non-diabetic patients. New-onset post-transplant diabetes doubled the likelihood of cancer development. Conclusions: Pre-transplant diabetes mellitus increases the risk of cancer after heart and/or lung transplantation. However, new-onset diabetes after transplantation is associated with a much greater cancer risk. This information is relevant for screening during follow-up.

## 1. Introduction

Cancer is a major source of morbidity and mortality in patients with solid organ transplantations [1]. Therefore, the identification of risk factors influencing the development of cancer after transplantation is important for follow-up. Numerous epidemiological studies have shown an association between diabetes and the risk of developing cancer in the normal population [2]. Diabetic patients are at significantly higher risk of developing common cancers, including pancreatic, liver, breast, colorectal, urinary tract, gastric, and female reproductive cancers [3,4,5,6,7,8,9]. The relative risk for developing cancer in diabetic patients ranges from 1.7 to 2.5 for liver, pancreatic, and endometrial cancers, and from 1.2 to 1.7 for breast, colon, and bladder cancers [3,4,5,6,7]. It is also known that, compared with the general population, recipients of kidney, liver, heart, or lung transplants have an increased risk for diverse cancers [10]. Furthermore, recipients of both heart and lung transplantation have significantly higher risk of any de novo cancer compared to recipients of other solid organs (e.g., liver) [11]. Thus, it is well conceivable that diabetes might also be a risk factor for cancer in the group of heart and lung transplant recipients.

We hypothesized that patients with diabetes after heart and/or lung transplantation are at greater risk of developing cancer compared to non-diabetic transplant patients. We tested our hypothesis in a retrospective cohort type study.

## 2. Materials and Methods

We retrospectively reviewed the medical records and our institutional database of all 353 patients post-heart and/or -lung transplantation who were transplanted in our center over a period of 22 years (between October 1999 and June 2021). The institutional ethics review committee approved the study protocol (reference number: 4927–09/16). Because of the slow nature of cancer occurrence, patients transplanted, lost to follow-up, or with survival of less than 180 days after transplantation were excluded from the main analysis (*n* = 87). The remaining 266 patients were divided into patients who had preoperative DM *(n* = 88) or developed it during follow-up *(n* = 40) and patients without DM (*n* = 138). To exclude a possible selection bias, an additional analysis of cancer occurrence in all 353 patients transplanted in our department (irrespective of their postoperative survival) divided by their diabetic status at the time of transplantation was also performed.

### 2.1. Pre-Transplantation Cancer Screening

Cancer was routinely ruled out in all patients before transplantation. The detailed pre-transplant screening protocol is shown in Appendix A.

### 2.2. Follow-Up

All patients are routinely followed in our department. After the initial hospital stay, patients are seen on a weekly and later monthly basis during the first 7 to 9 months after transplantation and, after 1.5 years, 3 times per year. A detailed overview of the follow-up protocol is shown in Appendix A.

### 2.3. Immunosuppressive Regimen

All patients transplanted in our department receive uniform standard immunosuppression therapy according to predefined standard operating procedures. A detailed overview of the immunosuppression regimen and the standard plasma levels are shown in Appendix A.

### 2.4. Statistical Analysis

Comparisons between the groups were performed using the Mann–Whitney U test for continuous and non-normally distributed variables; the data in each group are summarized by median and interquartile range (IQR). For continuous and normally distributed values, Student’s t-test was used, and mean ± standard deviation (SD) is provided for each group. Categorial data were compared by a chi-squared test or Fisher’s exact test. Survival probabilities were calculated using the Kaplan–Meier method and compared by a log-rank test. The significance level was set at *p* ≤ 0.05. All statistical analyses were performed using R version 3.6.2 (R Foundation for Statistical Computing, Vienna, Austria) and Stata version 17.0 (StataCorp, College Station, TX, USA).

Two methods were used to measure the effect of diabetes and new-onset diabetes on cancer after transplantation: a backward elimination logistic regression and a matching method using regression-adjusted inverse probability weights (IPWRA). The multivariable logistic regression model for cancer included the predictors age, gender, new-onset diabetes, previous diabetes, prednisolone therapy, mycophenolate, tacrolimus, and ciclosporin. The IPWRA considered mycophenolate therapy and age as the matching variables, since the two groups showed significant differences. As the IPWRA uses double regression for the outcome and the exposure variable, we could calculate an unbiased average effect of diabetes on cancer after transplantation.

## 3. Results

### 3.1. Patient’s Demographics

Table 1 shows patient demographic data from the entire and the weighted cohorts. After weighting, the sum of weights was 132.5 for the diabetic patients and 131.5 for the non-diabetic group.

Most patients were around 53 years of age in the diabetic group and 51 in the non-diabetic and were male. The mean follow-up period after transplantation was 8.15 ± 5.74 years for all patients included in the analysis. Both groups consisted mainly of heart transplant patients, followed by double lung transplants, single lungs, and a small number of other organ combinations.

The diabetic patients were slightly older and received more mycophenolate and less prednisolone than non-diabetic patients. There were no significant differences regarding the type of organ transplantation (heart, lung, or heart-lung) between groups. After inverse-probability-weighted regression adjustment, all baseline characteristics showed no significant differences between groups for the matching variables age and mycophenolate and showed a good balance, as shown in Table 1 and Appendix A. The standardized mean difference for all weighted observations was <0.1.

### 3.2. Incidence of Tumor/Neoplasm

Figure 1 shows the percentage of tumor/neoplasm developed in both groups during the whole observation period. The rate of malignancies detected during follow-up was almost twice as high in the diabetic group in the entire cohort (30.33 vs. 15.97%, *p* = 0.005—Figure 1A) and in the matched population (31.9 vs. 16.1%, *p* < 0.001—Figure 1B).

### 3.3. Type of Tumor/Neoplasm

Figure 2 shows the distribution of types of tumor/neoplasm developed in both groups during the whole observation period. The diabetic patients showed a significantly higher incidence of solid tumors in comparison with the non-diabetic group in the entire cohort (*p* = 0.042—Figure 2A) and close to significant in the matched population (*p* = 0.09—Figure 2B). There was no significant difference regarding the incidence of skin or myeloproliferative tumor/neoplasm.

### 3.4. Survival and Tumor-Free Survival

Figure 3A,B shows Kaplan-Meyer analyses displaying tumor-free survival rates in both groups during the whole observation period. The tumor-free survival differences did not reach statistical significance.

There was no overall survival difference between the pre-transplant diabetes patients and the ones without diabetes at the time of transplantation.

### 3.5. Role of Diabetes Onset and Exposure Effect

Table 2 shows the average exposure effect of diabetes on cancer after transplantation through inverse probability weights. After double regression adjustment for both the outcome (post-transplantation cancer) and the exposure variable (diabetes), it was shown that the effect of diabetes caused a 13% increase in cancer rate when compared to non-diabetic patients (95% confidence interval: 2.7–24.1%, *p* = 0.014). After backward elimination, it was shown that new-onset post-transplant diabetes and age were highly associated with post-transplantation cancer (Table 3).

## 4. Discussion

We demonstrate in this manuscript that pre-transplant diabetes mellitus mildly increases the risk of cancer after heart and/or lung transplantation. However, new-onset diabetes after transplantation is associated with a much greater cancer risk. This information is relevant for screening during follow-up and patient information.

De novo malignancies are a major source of morbidity and mortality in solid organ transplant recipients, and, probably due to the increased immunosuppressive therapy requirements, heart and/or lung transplant patients are at increased risk (e.g., four-fold compared to recipients of kidney transplantation) [12,13,14,15]. An analysis of more than 17,000 heart transplant patients from the International Society for Heart and Lung Transplantation Registry showed that more than 10% of adult heart recipients developed de novo malignancy between years 1 and 5 after transplantation [16]. The incidence of post-transplant de novo solid malignancy increased temporally, and older recipients and patients who underwent heart transplantation recently had a higher risk [16]. For lung transplant recipients, the International Society for Heart and Lung Transplantation report showed that malignancies are the second most common cause of death five to ten years out from transplantation (17.3%) and for patients who were at more than 10 years after the procedure (17.9%) [17].

Furthermore, the prevalence of diabetes mellitus is increasing worldwide, also affecting lung and heart transplant recipients [18]. An analysis of 108,034 heart transplant patients showed that the incidence of diabetes mellitus in the recipients divided by transplant period rose significantly (from 16.7% in the period 1992–2000 to 27% in 2010–2018) [18]. A similar increase in incidence was observed for lung transplant recipients [19]. A recent report from the International Thoracic Organ Transplant Registry showed that, in lung transplant recipients, the incidence of diabetes rose from 6.1% during the 1992–2000 period to 21.6% during the 2010–2018 period [19].

To the best of our knowledge, our analysis is the first to address the role of diabetes mellitus in cancer occurrence in thoracic organ transplant recipients. Although other investigations of malignancies after heart transplantation support our findings, they never commented on or addressed the role of diabetes in cancer occurrence in detail. For example, Rivinius et al. showed, in an analysis of de novo malignancies from the Heidelberg Registry for Heart Transplantation, that, among other factors, diabetes mellitus is also an independent risk factor for cancer after heart transplantation [15]. The authors did not explore or comment further on that relationship. In this line of argumentation, our findings might be further indirectly supported by the fact that some antidiabetic therapies might reduce cancer risk after transplantation. Peled et al. showed that metformin therapy is independently associated with a significant (90%) reduction in the risk of malignancy after heart transplantation [20]. Their finding has been confirmed recently by Bedanova et al., who also showed a significantly lower incidence of malignancies in metformin-treated patients and slightly better overall survival in their cohort of heart-transplanted patients [21].

One important aspect of our study is the finding that new-onset post-transplant diabetes is independently and highly associated with post-transplantation cancer. The dimension (plus 100% risk) was much greater than that of pre-transplant diabetes (plus 13% risk). Therefore, a pre-transplant exclusion of diabetes patients is unlikely to solve the problem, as it is currently not possible to predict which non-diabetic patient might develop new-onset post-transplant diabetes mellitus and who may, therefore, be at higher risk of cancer development.

Despite this risk, patients with diabetes and end-stage heart or lung diseases benefit significantly from transplantation, which often is the only survival option for them [22]. Nevertheless, our findings show that patients who are or become diabetic are at greater risk of cancer after heart and lung transplantation. It, therefore, appears reasonable to monitor them carefully, including frequent post-transplant cancer screenings. It is also reasonable to suggest careful treatment of diabetes, because evidence suggests that antidiabetic treatment may reduce this risk.

In addition, the fact that diabetes patients experience higher cancer occurrence is relevant, as recent studies showed that, in “chronic” cancer survivors, quality of life is impaired 2 to 26 years after cancer diagnosis [23]. Furthermore, the recent development of mRNA cancer vaccines (there are already clinical trials being conducted, e.g., NCT04526899) gives hope of significant advancements in this field and in potential upcoming cancer prevention and vaccination options. Therefore, our analysis contributes to defining risk factors connected with higher cancer occurrence in transplant patients and might have potential importance for selecting candidates for future anti-cancer vaccination.

### Limitations

This work has the intrinsic limitations of observational single-center studies. However, in contrast to multicenter studies, all patients received identical immunosuppressive therapy defined in standard operating procedures. Nevertheless, a bias may remain, because residual confounders were not considered. For this reason, we used two methods to measure the effect of diabetes on cancer after transplantation: a backward elimination logistic regression and a matching method using regression-adjusted inverse probability weights.

## 5. Conclusions

Pre-transplant diabetes mellitus mildly increases the risk of cancer after heart and/or lung transplantation. However, new-onset diabetes after transplantation is associated with a much greater cancer risk. This information is relevant for screening during follow-up and patient information.

## Figures and Tables

**Figure 1 jcm-11-04127-f001:**
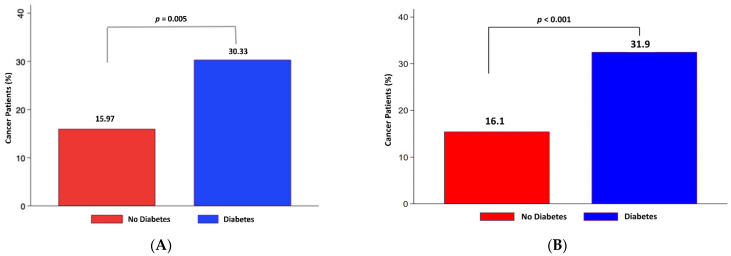
Cancer prevalence in diabetic and non-diabetic lung and/or heart transplant patients. (**A**) Bar chart showing the unadjusted percentage of patients with cancer without (red) and with diabetes mellitus (blue). The percentage of patients in relation to the total number of patients is indicated on the top of each column. (**B**) Bar chart showing the adjusted percentage of patients with cancer without (red) and with diabetes mellitus (blue). The percentage of patients in relation to the total number of patients is indicated on the top of each column.

**Figure 2 jcm-11-04127-f002:**
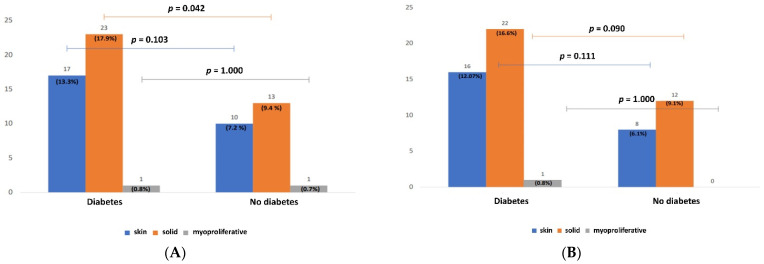
Distribution of cancer types among diabetic and non-diabetic lung and/or heart transplant patients. (**A**) Bar chart showing the unadjusted distribution of the different types of tumor/neoplasm (skin, solid, or myoproliferative—respectively indicated in blue, orange, and gray) in the groups with and without diabetes. The number of patients and the percentage of patients in relation to the total number of patients are indicated on the tops of the columns. (**B**) Bar chart showing the adjusted distribution of the different types of tumor/neoplasm (skin, solid, or myoproliferative—respectively indicated in blue, orange, and gray) in the groups with and without diabetes. The number of patients and the percentage of patients in relation to the total number of patients are indicated on the tops of the columns.

**Figure 3 jcm-11-04127-f003:**
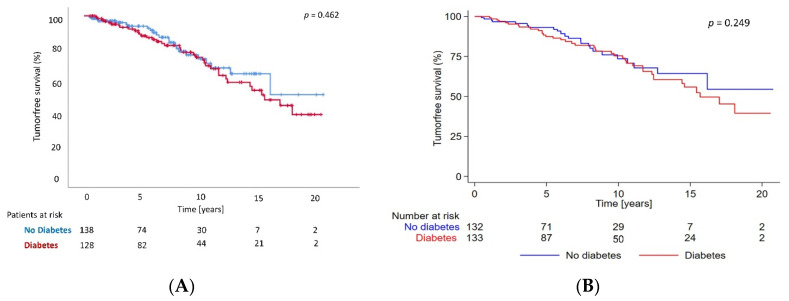
Tumor-free survival. (**A**)Tumor-free survival after 20 years of follow-up according to the presence or absence of diabetes in the unadjusted population. The numbers in the bottom part of the figure are the numbers of patients at risk. (**B**) Tumor-free survival after 20 years of follow-up according to the presence or absence of diabetes in the adjusted population. The numbers in the bottom part of the figure are the weights of at-risk patients.

**Table 1 jcm-11-04127-t001:** Patient Demographics.

	Before Weighting			After Weighting		
	Diabetes *	No Diabetes		Diabetes *	No Diabetes	
	(*n* = 128)	(*n* = 138)	*p*	(SW = 132.5)	(SW = 131.5)	*p*
Age (y), average ± SD	53.2 ± 11.0	50.9 ± 11.9	0.057	51.5 ± 12.4	51.9 ± 11.4	0.786
Male sex, n, (%)	98 (76.6)	94 (68.1)	0.125	101.2 (76.42)	89.4 (68.4)	0.148
Diabetes at time of transplantation n, (%)	88 (68.75)					
**Type of diabetes**						
Type 1, *n*, (%)	5 (3.9)	0				
Type 2, *n*, (%)	117 (91.4)	0				
NODAT, *n*, (%) ^1^	6 (4.7)	0				
**Diabetes therapy**						
Diet, *n*, (%)	38 (29.7)	0				
Oral therapy, *n*, (%)	26 (20.3)	0				
Insulin-dependent, *n*, (%)	64 (50)	0				
**Type of transplantation**						
Heart, *n*, (%)	76 (59.3)	68 (49.3)	0.099	77.3 (58.4)	66.4 (50.5)	0.199
Lung on pump, *n*, (%)	5 (3.9)	11 (8.0)	0.164	5.8 (4.4)	9.5 (7.2)	0.326
Lung off pump, *n*, (%)	42 (32.8)	53 (38.4)	0.341	43.7 (33.3)	49.9 (38.4)	0.381
Heart-lung, *n*, (%)	5 (3.9)	6 (4.3)	0.857	5.2 (3.9)	5.1 (3.5)	0.966
**Immunosuppression**						
Tacrolimus, *n*, (%)	97 (75.8)	100 (72.5)	0.537	100.7 (76.2)	96.0 (72.9)	0.540
Mycophenolate, *n*, (%)	118 (92.2)	115 (83.3)	0.029	115.3 (87.3)	115.7 (88.1)	0.844
Prednisolone, *n*, (%)	66 (51.6)	85 (61.6)	0.099	68.9 (52.0)	81.9 (62.3)	0.090

SD = standard deviation, SW = sum of weights. * Including all patients with diabetes mellitus type 1, type 2, and new-onset diabetes after transplantation at time of data collection. ^1^ New-onset diabetes after transplantation.

**Table 2 jcm-11-04127-t002:** Average Exposure Effect of Diabetes using Inverse-Probability-Weighted Regression Adjustment.

Cancer after Transplantation	Coefficient	*p*-Value	95% Confidence Interval
**Average Exposure Effect**			
Diabetes versus No diabetes	0.134	0.014	0.027–0.241
**Mean Potential Outcome**			
No diabetes	0.165	0.001	0.097–0.233

**Table 3 jcm-11-04127-t003:** Results of the multivariable logistic regression analysis with backward elimination.

Cancer after Transplantation	Odds Ratio	*p*-Value	95% Confidence Interval
New-onset post-transplant diabetes	2.58	0.025	1.121–3.250
Age (in years)	1.03	0.03	1.001–1.006

## Data Availability

Not applicable.

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
