# Peer review of "Diabetes Is an Independent Risk Factor for Cancer after Heart and/or Lung Transplantation"

_jcm, 2022, doi:10.3390/jcm11144127_

Round 1

Reviewer 1 Report

I want to thank the authors for implementing the suggested changes. It strengthened the manuscript significantly.

Some minor suggestions to the current version:

1. The first sentence of the abstract is not understandable and needs revision. Please proofread the entire abstract, as some more sentences need syntax correction.

2. Correct the heading of Table 1. There is a discrepancy between the marked and unmarked versions of the manuscript.

3. I suggest keeping the same data presentation format in Figures 1 and 2 (# of patients or % on the top of the columns) 

4. Figure 3B. The # of patients at risk is not correct. Please check and provide the valid integer values.

Author Response

jcm-1815506

Diabetes is an Independent Risk Factor for Cancer after Heart and/or Lung Transplantation

Response to reviewers

We thank the reviewers for their favorable comments. We have revised the manuscript to address all comments. Please find below our point-by-point response.

Reviewer #1 Comments:

I want to thank the authors for implementing the suggested changes. It strengthened the

manuscript significantly.

Some minor suggestions to the current version:

  1. The first sentence of the abstract is not understandable and needs revision. Please proofread the entire abstract, as some more sentences need syntax correction.
  2. Correct the heading of Table 1. There is a discrepancy between the marked and unmarked versions of the manuscript.
  3. I suggest keeping the same data presentation format in Figures 1 and 2 (# of patients or % on the top of the columns)
  4. Figure 3B. The # of patients at risk is not correct. Please check and provide the valid integer values.

Response:

We thank the Reviewer for the helpful and favorable comments. We have addressed al mentioned points.

  1. We have corrected and proofread the entire abstract.
  2. We have corrected the heading of Table 1.
  3. We have included the % for Figure 2 on the top of the columns as recommended.
  4. We have corrected the # of patients at risk.

Reviewer 2 Report

The authors have responded to all my suggestions. The only thing missing is to adjust by including the covariates not completely adjusted in the final models, which will limit their impact.

Author Response

jcm-1815506

Diabetes is an Independent Risk Factor for Cancer after Heart and/or Lung Transplantation

Response to reviewers

We thank the reviewers for their favorable comments. We have revised the manuscript to address all comments. Please find below our point-by-point response.

Reviewer #2 Comments:

The authors have responded to all my suggestions. The only thing missing is to adjust by including the covariates not completely adjusted in the final models, which will limit their impact.

Response:

We thank the Reviewer for his favorable comments. As already described in our response letter we decided to apply inverse probability weights regression-adjustment (IPWRA) instead of propensity score matching (PSM), because PSM has methodological weaknesses compared to IPWRA (e.g. discarding unmatched observations and the bias of outcome estimates due to misspecification of the propensity score model). With IPWRA we already achieved a balanced distribution of all baseline variables (see table 1), so there is no need to adjust for any of these covariates.

This manuscript is a resubmission of an earlier submission. The following is a list of the peer review reports and author responses from that submission.

Round 1

Reviewer 1 Report

The work presented is interesting. However, to avoid residual difference bias, a double-adjustment in propensity score matching analysis should be applied to the main results.
 For the validation of the propensity score model, it will also be necessary to propose the set of tests (in particular graphs of the residuals....) to illustrate the differences or the equivalence of the parameters between the groups. 

Reviewer 2 Report

In the study of Kirov and colleagues the authors analyzed the effect of diabetes on the incidence of cancer after heart/lung transplantation. In the cohort of 266 patients, they found that the incidence of cancer was significantly higher in the diabetic group than in the no-diabetics (28.9 vs 16.7%). This result was confirmed in the propensity matched population of 240 patients. Moreover, the authors found that the most common type of cancer in diabetic patients after transplantation were solid tumors. However, based on the results presented It looks the diabetes had no effect on the tumor free survival in the studied groups.

I would like to congratulate the authors on the interesting and important study. However, there are several major issues that need to be addressed:

1.     The authors run the analysis on the entire cohort and matched population but the matching needs further clarification.

As the authors stated, they matched among other parameters on the survival time. It looks like the authors matched based on the outcome variable. Why did they decide to do so and did not adjust for follow up time instead?  The overall survival time is in fact not provided in the manuscript. The only difference the reader can found in this regard is the tumor free survival. If that was the parameter used that is not an acceptable approach as this was the outcome analyzed.

Second, it is not clear how the matching was performed. Having the same number of patients in the diabetic group before and after matching suggest that what was only done was the exclusion of some non-diabetic patients. That is not an acceptable approach. The authors should aim to look at the matched pairs or follow closely pre specified matching protocol.

2.     In the supplementary Figure 1 we can see significantly lower survival in the non- diabetic population. This is very important result but it is neither presented nor commented in the main manuscript. How do the authors can explain this difference in the context of their analysis? I suggest to perform the multivariate analysis (Cox) to see what is responsible for that. That should also be an approach taken to analyze other results in the study.

The mere fact of more cancer occurrence in the diabetic population is not that important if the authors cannot show any relation of that fact with any outcome (survival, QOL etc). Especially as clearly presented in the discussion section it is known that malignancies are a major source of mortality and morbidity in transplant patients. The authors should analyze that in the current study as well.

3.     The authors in Figure 3 show no difference in tumor free survival but state otherwise in the text of both abstract and the main manuscript. The null hypothesis of such survival analysis and log-rank test is that there is no difference in survival between both groups. If the authors cannot reject it, they are not allowed to state that: “The diabetic patients had lower tumor-free survival in comparison with the non-diabetic group”.

4.     The authors provide the incidence of type 3 diabetes explaining it as an “safely steroid induced”. Type 3 diabetes is not a widely used term and is most commonly referred as the alternative name for Alzheimer’s disease, as, in some cases, Alzheimer’s can be linked to insulin resistance. I am not sure if this is what the authors wanted to show. Regardless ‘type 3 diabetes’ is still a controversial term.

The authors should also double check the spelling across the manuscript and supplementary material (especially Supplementary Table 4).

Again, I would like to congratulate the authors and I hope they can rerun some of their analysis to strengthen the manuscript and make it more impactful.